# Synthesis of Arylene Ether-Type Hyperbranched Poly(triphenylamine) for Lithium Battery Cathodes

**DOI:** 10.3390/ma14247885

**Published:** 2021-12-20

**Authors:** Inah Kang, Taewoong Lee, Young Rok Yoon, Jee Woo Kim, Byung-Kwon Kim, Jinhee Lee, Jin Hong Lee, Sang Youl Kim

**Affiliations:** 1Department of Chemistry, Korea Advanced Institute of Science and Technology (KAIST), Daejeon 34141, Korea; gk0784@kaist.ac.kr (I.K.); yoonyr396@kaist.ac.kr (Y.R.Y.); 2School of Chemical Engineering, Pusan National University, Busan 46421, Korea; wmfrlwk12@naver.com; 3Department of Chemistry and Nanoscience, Ewha Womans University, Seoul 03760, Korea; jw1603052@gmail.com (J.W.K.); kimb@ewha.ac.kr (B.-K.K.); 4Reliability Assessment Center for Chemical Materials, Korea Research Institute of Chemical Technology (KRICT), Daejeon 34114, Korea

**Keywords:** triphenylamine, S_N_Ar reaction, hyperbranched poly(triphenylamine), lithium-ion battery, polymer cathode

## Abstract

We synthesized a new poly(triphenylamine), having a hyperbranched structure, and employed it in lithium-ion batteries as an organic cathode material. Two types of monomers were prepared with hydroxyl groups and nitro leaving groups, activated by a trifluoromethyl substituent, and then polymerized via the nucleophilic aromatic substitution reaction. The reactivity of the monomers differed depending on the number of hydroxyl groups and the A_2_B type monomer with one hydroxyl group successfully produced poly(triphenylamine). Based on thermal, optical, and electrochemical analyses, a composite poly(triphenylamine) electrode was made. The electrochemical performance investigations confirmed that the lithium-ion batteries, fabricated with the poly(triphenylamine)-based cathodes, had reasonable specific capacity values and stable cycling performance, suggesting the potential of this hyperbranched polymer in cathode materials for lithium-ion batteries.

## 1. Introduction

Organic charge-transporting materials have been extensively applied in opto-electronic devices [1,2]. They transport charge carriers, i.e., holes and electrons, as well as generate charge carriers in response to external stimuli [3]. The electronic structure and processability of organic charge-transport materials can be tuned by chemical modification, and opto-electronic devices can be fabricated with organic charge-transporting polymeric materials by using low-cost approaches, such as solution processes [4,5,6,7]. Various types of conductive polymers with π-conjugation, such as polythiophene, polypyrrole, and polyaniline, have been synthesized and have shown excellent performance in opto-electronic devices [8,9,10,11].

The efficient charge transfer abilities of conductive polymers have also been utilized in organic cathode materials for lithium-ion batteries (LIBs) [12,13,14,15,16,17]. By replacing the redox-active transition metals, conventionally used in cathodes, with conductive polymers, it has been possible to fabricate a system that has high oxidation stability in air and an intrinsically high surface area for small reactants, leading to high capacitive behaviors [18,19,20,21]. Additionally, the conductive pathways of the long-conjugated polymer chains can facilitate redox electron transfer and reduce electrode/electrolyte interface resistance, allowing for a high-rate capability. However, because the charge centers are not separated electronically, they strongly affect each other, and thus, only a limited number of active moieties are involved during charging/discharging, resulting in an increase in cell polarization. For these reasons, recent studies have focused more on polymers with a nonconductive backbone that bear electroactive pendant groups and organic radicals because of their localized redox sites [22,23].

Poly(triphenylamine) is one of the representative conductive polymers. It transports positive charges, via a stable radical cation, on its nitrogen atom [24]. The propeller shaped triphenylamine provides a large surface area that is highly favorable to ion transfer [25,26,27,28,29]. It also possesses a partially conjugated system, attributed to the limited π-conjugation of the triphenylamine unit at its neutral state [30]. Such properties create an effect, similar to those in existing polymers, which contains electroactive pendant groups, when applied to LIBs. Indeed, poly(triphenylamine) has been applied to LIBs, accompanied by various structures with linear, hyperbranched, or highly cross-linked morphology. Most of them have shown promising performance and high reproducibility [31,32,33,34,35,36,37,38,39,40,41,42,43]. Among them, hyperbranched polymers with a large surface area enhance LIB performance compared to linear type polymers. Furthermore, a propeller shaped structure of triphenylamine effectively reduces the π-π stacking between the polymer chains. Therefore, introducing the triphenylamine unit into a highly branched structure can provide enhanced electrolyte diffusion and eventually improve the LIB performance compared to the linear type polymers. Unlike the highly crosslinked polymers, in addition, hyperbranched polymers have good solubility in organic solvents, which allows them a solution-based fabrication [44,45,46,47].

Here, we report a new type of hyperbranched poly(triphenylamine) for a LIB cathode. The poly(triphenylamine) synthesized in this study consists of triphenylamine and arylene ether units that are expected to provide high thermal and dimensional stability [48,49]. There are several methods for producing the hyperbranched polymers, for example, by reacting a single monomer with multifunctional reaction sites (AB*_n_* or latent AB*_n_* type) or multiple monomers simultaneously [44]. We firstly prepared two kinds of monomer, AB_2_ and A_2_B, which contained hydroxyl groups and nitro leaving groups, activated by a trifluoromethyl group at the ortho position, as highlighted in Figure 1. However, only the A_2_B type monomer produces the desired hyperbranched polymer due to its reactivity. Subsequently, the polymer was utilized as a cathode material for LIBs. We investigated the electrochemical performances of cells fabricated with the poly(triphenylamine)-based cathode via cyclic voltammetry, rate capability, and long-term cycling stability evaluation.

## 2. Materials and Methods

### 2.1. Materials

Tris (dibenzylideneacetone) dipalladium (0) (Pd_2_(dba)_3_), 2,2′-bis (diphenylphosphino)-1,1′-binaphthalene (BINAP), sodium tert-butoxide (NaO*t*Bu), anhydrous dimethylacetamide (DMAc), anhydrous toluene, pyridine hydrochloride, ferrocene, and tetrabutylammonium hexafluorophosphate (TBAHFP) were purchased from Sigma Aldrich. *p*-Anisidine, 4-bromoanisole, and 5-bromo-2-nitrobenzotrifluoride were purchased from TCI. 1,2-Dimethoxyethane (DME), potassium phosphate (K_3_PO_4_), acetone, and benzene were purchased from Junsei Chemical. Acetonitrile was purchased from Daejung. All commercially available reagent-grade chemicals were used without additional purification.

### 2.2. Material Characterization

^1^H and ^13^C NMR (nuclear magnetic resonance) spectra of synthesized materials were recorded on an Agilent Technologies DD2 600 spectrometer. Chemical shifts were expressed in ppm (part per million), with reference to the peaks of residual DMSO for ^1^H (2.49 ppm) and ^13^C (39.52 ppm). Mass spectra were taken with microTOF-Q II, and elemental analysis was obtained with FLASH 200 series. FT-IR (Fourier transform infrared spectroscopy) spectra were recorded on a Varian Cary5000. Molecular weights and molecular weight distributions of polymers were obtained by size exclusion chromatography (SEC). The SEC traces were taken using a Viscotek TDA302 device (Malvern, UK) that is equipped with a refractive index detector and three PL gel 10 μm MIXED-B columns, with tetrahydrofuran as an eluent, at 36 °C. The weight and number of the average molecular weights of the polymers were calibrated compared to linear polystyrene standards. TGA (Thermogravimetry analysis) and DSC (differential scanning calorimetry) were conducted on a TA Instruments TGA Q500 and a DSC Q20, respectively. The TGA and DSC measurements were performed, under a nitrogen atmosphere, at a heating rate of 10 °C/min. *T*_g_ values were taken from the 2nd heating scan of the DSC thermograms after cooling from 300 °C to 0 °C. UV/Vis absorption spectra were obtained on Thermo Fisher Scientific Nicolet 5700 in acetone. X-ray photoelectron spectroscopy (XPS) measurements were performed with a Kratos AXIS Supra (Manchester, UK). Scanning electron microscope was used to observe the morphology of a polymeric cathode using a Zeiss SUPRA 25 (Oberkochen, Germany). The electrochemical experiments were carried out using a CH instrument (CHI 630E potentiostat) equipped with a three-electrode system. Pt wire was used as a counter electrode, and silver/silver nitrate (Ag/AgNO_3_) was used as a reference electrode. A 2 mm diameter gold disk was employed as a working electrode. All potentials were referenced against the Ag/Ag^+^ redox couple. Values of E_1/2_ of the ferrocene/ferrocenium ion couple were 0.0915 V in this experimental condition.

### 2.3. Synthesis of the AB_2_ Type Monomer

#### 2.3.1. Bis(4-methoxyphenyl)amine (1)

To a two-necked round-bottomed flask (RBF) having a reflux condenser, *p*-anisidine (5.419 g, 33 mmol), 4-bromoanisole (7.482 g, 40 mmol), Pd_2_(dba)_3_ (0.183 g, 0.2 mmol), BINAP (0.373 g, 0.6 mmol), NaO*t*Bu (5.382 g, 56 mmol), and anhydrous toluene (60 mL) were added, and the reaction mixture was stirred at 120 °C for 12 h under nitrogen gas flow. The reaction solution was allowed to cool to room temperature, diluted with ethyl acetate, and washed with water. After drying of the product with magnesium sulfate and removal of the solvent, the crude product was purified by column chromatography on silica gel (ethyl acetate: hexane = 1:10). The product was recrystallized from ethyl acetate and hexane mixed solvent to give 6.170 g (67 %) of a white solid 1.

^1^H NMR (DMSO-*d*6, 600 MHz, ppm): 7.50 (s, 1H), 6.92 (d, *J* = 8,9 Hz, 4H), 6.80(d, *J* = 8.9 Hz, 4H), 3.67 (s, 6H). ^13^C NMR (DMSO-*d*6, 150 MHz, ppm): 152.83, 138.05, 118.04, 114.54, 55.20. Anal. Calcd. for C_14_H_15_NO_2_: C, 73.34; H, 6.59; N, 6.11; O, 13.96. Found: C, 72.76; H, 6.43; N, 5.63; O, 13.13. ESI-MS: N/A (Calcd. for [M]: 229.11).

#### 2.3.2. Bis(4-methoxyphenyl)-(4-nitro-3-trifluoromethylphenyl)amine (2)

While already having a reflux condenser, 1 (3.837 g, 16.74 mmol), 5-bromo-2-nitrobenzotrifluoride (4.971 g, 18.41 mmol), Pd_2_(dba)_3_ (0.077 g, 0.084 mmol), BINAP (0.208 g, 0.335 mmol), K_3_PO_4_ (4.973 g, 23.430 mmol), and DME (50 mL) were added to a two-necked RBF. The reaction mixture was stirred at 100 °C for 24 h under nitrogen atmosphere. The reaction solution was allowed to reach room temperature, diluted with ethyl acetate, and washed with water. After the drying of the product with magnesium sulfate and removing of the solvent, the crude product was purified by column chromatography on silica gel (ethyl acetate:hexane = 1:3). The final product was recrystallized from ethyl acetate/hexane mixed solvent to afford 6.747 g (96% yield) of an orange solid 2.

^1^H NMR (DMSO-*d*6, 600 MHz, ppm): 8.04 (d, *J* = 9.2 Hz, 1H), 7.16 (dd, *J* = 167.1, 8.9 Hz, 8H), 6.88 (d, *J* = 2.7 Hz, 1H), 6.80 (dd, *J* = 9.2, 2.7 Hz, 1H), 3.76 (s, 6H). ^13^C NMR (DMSO-*d*6, 150 MHz, ppm): 157.99, 152.85, 136.79, 136.06, 129.07, 124.25 (q, *J*_C-F_ = 32.4 Hz), 122.17 (q, *J*_C-F_ = 273.0 Hz), 116.77, 115.60, 112.67 (q, *J*_C-F_ = 6.5 Hz), 55.34. Anal. Calcd. for C_21_H_17_F_3_N_2_O_4_: C, 60.29; H, 4.10; F, 13.62; N, 6.70; O, 15.30. Found: C, 60.04; H, 3.79; N, 6.62; O, 15.59. ESI-MS: 441.10 for [M+Na]^+^ (Calcd. for [M]: 418.11).

### 2.4. Synthesis of the A_2_B Type Monomer

#### (4-Methoxyphenyl)-bis(4-nitro-3-trifluoromethylphenyl)amine (3)

While already having a reflux condenser, 5-bromo-2-nitrobenzotrifluoride (8.910 g, 33 mmol), *p*-anisidine (1.847 g, 15 mmol), Pd_2_(dba)_3_ (0.137 g, 0.15 mmol), BINAP (0.373 g, 0.6 mmol), K_3_PO_4_ (8.915 g, 42 mmol), and DME (70 mL) were added to a two-necked RBF and stirred at 100 °C for 36 h in nitrogen atmosphere. The reaction solution was cooled to room temperature, diluted with ethyl acetate, and washed with water. After the drying of the product with magnesium sulfate and removing of the solvent, the crude product was purified by column chromatography on silica gel (ethyl acetate:hexane = 1:5). The product was recrystallized from ethyl acetate/hexane mixed solvent to give 7.064 g (94% yield) of an orange solid.

^1^H NMR (DMSO-*d*6, 600 MHz, ppm): 8.14 (d, *J* = 8.9 Hz, 2H), 7.49 (m, 4H), 7.21 (dd, *J* = 149.8, 8.6 Hz, 4H), 3.80 (s, 3H). ^13^C NMR (DMSO-*d*6, 150 MHz, ppm): 158.66, 150.01, 141.17, 136.01, 129.28, 128.39, 125.41, 123.90 (q, *J*_C-F_ = 33.4 Hz), 121.85 (q, *J*_C-F_ = 273.3 Hz), 120.30 (q, *J*_C-F_ = 5.8 Hz), 116.05, 55.39. Anal. Calcd. for C_21_H_13_F_6_N_3_O_5_: C, 50.31; H, 2.61; F, 22.74; N, 8.38; O, 15.96. Found: C, 48.79; H, 2.08; N, 7.21; O, 17.22. ESI-MS: 524.07 for [M+Na]^+^ (Calcd. for [M]: 501.08).

### 2.5. Deprotection of Methoxy Groups

A 250 mL RBF equipped with a condenser was charged with 1 mmol of the monomer and 33 mmol of pyridine hydrochloride was heated to 210 °C for 1 h. The reaction solution was cooled to room temperature and then diluted with water. After extraction of the product with ethyl acetate, it was washed with water. After drying of the product with magnesium sulfate, the excess solvent was removed.

#### 2.5.1. Bis(4-hydroxyphenyl)-(4-nitro-3-trifluoromethylphenyl)amine (4)

Deprotection of monomer 2 (5.076 g, 12.14 mmol) was carried out with pyridine hydrochloride. The product was purified by column chromatography in silica gel (ethyl acetate: hexane = 1:2) and recrystallized from ethyl cyclohexane/methylene chloride to afford 4.010 g (79% yield) of a yellow solid.

^1^H NMR (DMSO-*d*6, 600 MHz, ppm): 9.72 (s, 2H), 8.02 (d, *J* = 9.3 Hz, 1H), 7.01 (dd, *J* = 193.8, 8.8 Hz, 6H), 6.82 (s, 1H), 6.74 (d, *J* = 9.3 Hz, 1H). ^13^C NMR (DMSO-*d*6, 150 MHz, ppm): 156.46, 153.24, 135.59, 135.36, 129.16, 128.61, 124.33 (q, *J*_C-F_ = 32.2 Hz), 122.27 (q, *J*_C-F_ = 273.0 Hz), 116.92, 116.18, 112.28 (q, *J*_C-F_ = 6.4 Hz). Anal. Calcd. for C_19_H_13_F_3_N_2_O_4_: C, 58.47; H, 3.36; F, 14.60; N, 7.18; O, 16.40. Found: C, 55.05; H, 3.26; N, 6.73; O, 17.30. ESI-MS: 413.07 for [M+Na]^+^ (Calcd. for [M]: 390.08).

#### 2.5.2. 4-(Bis(4-nitro-3-trifluoromethylphenyl)amino)phenol (5)

Deprotection of monomer 3 (6.490 g, 12.95 mmol) was carried out with pyridine hydrochloride. The product was purified by column chromatography in silica gel (ethyl acetate: hexane = 1:3) and recrystallized from ethyl acetate/hexane to afford 5.385 g (83% yield) of an orange solid.

^1^H NMR (DMSO-*d*6, 600 MHz, ppm): 9.92 (s, 1H), 8.14 (d, *J* = 8.8 Hz, 2H), 7.49–7.45 (m, 4H), 7.04 (dd, J = 180.2, 8.7 Hz, 4H). ^13^C NMR (DMSO-*d*6, 150 MHz, ppm): 157.19, 150.06, 140.98, 134.41, 129.37, 128.40, 125.19, 123.80 (q, *J*_C-F_ = 33.1 Hz), 121.87 (q, *J*_C-F_ = 273.2 Hz), 120.11 (q, *J*_C-F_ = 5.6 Hz), 117.40. Anal. Calcd. for C_20_H_11_F_6_N_3_O_5_: C, 49.29; H, 2.28; F, 23.39; N, 8.62; O, 16.42. Found: C, 58.72; H, 3.85; N, 5.24; O, 18.46. ESI-MS: 510.05 for [M+Na]^+^ (Calcd. for [M]: 487.31).

### 2.6. Potassium Phenoxides Formation of Monomer

Monomer 4 or 5 (1 mmol) and 50 mL of methanol were charged into a 250 mL RBF. Potassium hydroxide (0.99 mmol) was dissolved in methanol (20 mL), and the solution was poured slowly into the flask at 0 °C. The mixture was stirred at room temperature for 1 h. Solvent was removed by a rotary evaporator, and then acetone (100 mL) was added to the flask. The reaction mixture was filtered and washed with dichloromethane. The mixture was dried in a vacuum oven to afford the product, quantitatively. The monomer potassium phenoxides prepared (designated as 4-K and 5-K) were characterized by ^1^H and ^13^C NMR spectroscopies.

4-K ^1^H NMR (DMSO-*d*6, 600 MHz, ppm): 8.01 (d, *J* = 9.4 Hz, 1H), 7.07 (d, *J* = 8.2 Hz, 4H), 6.81 (s, 1H), 6.74 (d, *J* = 8.3 Hz, 4H), 6.72 (s, 1H). ^13^C NMR (DMSO-*d*6, 150 MHz, ppm): 158.53, 153.51, 135.05, 133.91, 129.21, 128.35, 124.30 (q, J_C-F_ = 32.3 Hz), 122.28 (q, J_C-F_ = 272.8 Hz), 117.20, 115.78, 112.01 (q, J_C-F_ = 6.6 Hz).

5-K ^1^H NMR (DMSO-*d*6, 600 MHz, ppm): 8.12 (d, J = 9.4 Hz, 2H), 7.47 (d, J = 6.0 Hz, 4H), 6.87 (dd, J = 172.7, 8.2 Hz, 4H). ^13^C NMR (DMSO-*d*6, 150 MHz, ppm): 162.74, 150.30, 140.58, 130.39, 128.92, 128.38, 124.71, 123.63 (q, *J*_C-F_ = 33.2 Hz), 121.89 (q, *J*_C-F_ = 273.0 Hz), 119.83 (q, *J*_C-F_ = 5.9 Hz), 118.38.

### 2.7. Polymerization

A 25 mL of three-necked RBF, having a nitrogen inlet, Dean–Stark trap, mechanical stirrer, and a reflux condenser, was charged with monomer (1.5 g, 3.1 mmol), K_2_CO_3_ (0.7445 g, 5.4 mmol), anhydrous DMAc (5 mL), and benzene. The mixture was heated to 110 °C, and through the Dean–Stark trap, water was removed by azeotropic distillation of toluene for 3 h. Then, the temperature was increased to 160 °C. The reaction mixture was polymerized for 3 h. After the reaction mixture was cooled to room temperature, it was poured into water that was acidified with a small amount of acetic acid. The precipitated polymer was collected by filtration and further purified by the reprecipitation, of the DMAc solution of polymer, into methanol. The filtered polymer product was dried in a vacuum oven.

### 2.8. Electrochemical Characterization

The coin type CR2032 cells were fabricated in an argon-filled glove box to measure electrochemical performance. The electrode slurry was prepared at a weight ratio of 50 wt% of the as-prepared polymer, 10 wt% of polyvinylidene difluoride (PVDF), and 40 wt% of super P in 1-methyl-2-pyrrolidinone (Sigma Aldrich, >99.0 %, St. Louis, MO, USA) solution homogenously mixed by Thinky mixer and dried for 24 h at 60 °C in a vacuum oven. Lithium metal, as a counter electrode and PP separator (Celgard 2400), was used in fabrication of CR2032 (Seoul, Korea). Additionally, 1 M LiPF_6_ in 1:1 EC (ethylene carbonate)/DMC dimethyl carbonate was used as an electrolyte. Charge/discharge galvanostatic experiments were performed using an automatic battery cycler (WonATech, WBCS3000, Seoul, Korea) in the voltage range of 2.5–4.2 V (vs. Li/Li+). Rate capabilities were examined at different current densities from 20 mA/g to 500 mA/g.

## 3. Results

### 3.1. Synthesis and Characterization

In this study, we designed two types of triphenylamine monomer, A_2_B and AB_2_, to synthesize poly(triphenylamine)s with a hyperbranched structure. Each monomer includes a hydroxyl group (-OH) and a nitro group (-NO_2_) as a leaving group for the nucleophilic aromatic substitution (S_N_Ar) reaction: the AB_2_ type contains two hydroxyl groups and one nitro group, with an electron-withdrawing trifluoromethyl group at the ortho position. The A_2_B type, in contrast, has one hydroxyl group and two nitro groups, as described in Figure 1. During the reaction, the phenoxide groups attack the ipso carbon of the aromatic rings. Then the nitro-leaving groups that are activated with the trifluoromethyl groups are substituted to form the ether bonds, which results in hyperbranched poly(triphenylamine)s.

Figure 1 summarizes our synthetic route to the monomers. *p*-Anisidine and 4-bromoanisole underwent a palladium-catalyzed amination reaction to form a diphenylamine compound 1. The intermediate 1 was coupled with 5-bromo-2-nitrobenzotrifluoride, which produced a triphenylamine compound 2 with a methoxy-protected hydroxyl group. When the methoxy groups in 2 were deprotected with pyridine hydrochloride, an AB_2_ type monomer 4 was obtained. Meanwhile, coupling the 5-bromo-2-nitrobenzotrifluoride with *p*-anisidine afforded a methoxy-protected triphenylamine 3, and the same deprotection process gave an A_2_B type monomer 5. The chemical structure of the monomers was confirmed by ^1^H and ^13^C NMR, as shown in Appendix A.

Next, the polymerization of the two types of monomers was conducted, respectively, via the S_N_Ar reaction. We reacted monomers 4 and 5 in DMAc at 160 °C assisted by K_2_CO_3_ as a base catalyst, as shown in Figure 2. To prevent a nucleophilic attack by water, it was carried out under anhydrous conditions, preceded by the azeotrope. After 3 h, the reaction of 5 afforded the desired triphenylamine polymer (PTPA) with a 71% yield. On the other hand, interestingly, no polymeric product for 4 was obtained.

We posited that this occurred because of the reactivity of the monomers. According to former studies [50,51,52], the reactivity toward nucleophiles can be proved by NMR spectroscopy, by comparing the spectra of the phenoxide form that was exploited during the reaction. To do so, we prepared the potassium phenoxides of monomer 4 and 5, respectively (designated as 4-K and 5-K), and examined the ^13^C NMR peaks of the nitro-leaving group in 4-K and 5-K (Appendix A). Figure 2 clearly shows that the peak, corresponding to an ipso carbon of the nitro group in 5-K, was highly deshielded, while the one in 4-K shows an upfield shift. This large downfield shift in 5-K indicates a deficiency of electrons in the ipso carbons because of the electrons pulled by the nitro group. This helped to activate the nitro group as the leaving group and, consequently, encouraged the successive reaction. However, the nitro group in 4-K lacked the electron-withdrawing ability that was required for the reaction, and thus, polymerization did not occur. This result supports that the number of the hydroxyl group is related to the reactivity, and the A_2_B type monomer 5, which includes less hydroxyl groups, favored the S_N_Ar reaction.

To investigate further, a variety of spectroscopic techniques were used. In the ^1^H NMR spectra of PTPA (Figure 3), a series of peaks at 6.5–8.2 ppm, corresponding to the protons of the hyperbranched aromatic rings, revealed that PTPA had been synthesized successfully. Although the undesired displacement of nitro groups can occur during the polymerization [53,54,55,56] any related peaks were not observed in the spectrum, indicating there was no side reaction. The infrared spectra of 5 and PTPA, shown in Appendix A, also supported the conclusion of polymerization. Most of the hydroxyl groups in 5 were consumed, since the associated peak at 3434 cm^−1^ disappeared from the PTPA spectrum. Likewise, the peak at 1575 cm^−1^ corresponding to N-O stretching largely diminished but still remained after the polymerization. This indicates the elimination of nitro groups from 5 as well, yet a notable amount of nitro groups remained at the end of the hyperbranched polymer. In addition, the molecular weight of the PTPA was estimated by SEC using linear polystyrene standards, as shown in Appendix A. XPS spectrum of PTPA (Appendix A) also shows the corresponding peaks, for elements (C, N, O, F) present in the PTPA, synthesized in this study. These results accurately confirmed that the polymerization was successful.

TGA and DSC curves indicated the PTPA had excellent thermal properties. The 5% weight loss temperature (*T*_d5_) was 417 °C, and the glass transition temperature (*T*_g_) was observed at 202 °C, with no melting transition up to 300 °C. This suggests that PTPA within a cathode will not be transformed or damaged when the temperature is elevated, as summarized in Figure 4.

Next, we investigated optical properties from the UV/Vis absorption spectra of 5 and PTPA. As depicted in Figure 5a, the maximum absorption of 5 and PTPA appeared at 403 and 390 nm, respectively, which corresponds to the π–π * electron transition from the triphenylamine units. It was observed that the absorption became blue-shifted after the polymerization, which is attributed to the decrease in the number of electron-withdrawing nitro groups. The energy gap of the PTPA, which was calculated from the onset of absorption from the UV spectrum, was 2.73 eV.

To investigate its electronic structure, the electrochemical behavior of PTPA was recorded using cyclic voltammetry (CV). Figure 5b shows a clearly reversible CV curve of 0.1 mM for PTPA, immersed in 100 mM of TBAHFP acetonitrile solution, scanned up to 1.4 V of the potential. The onset of oxidation potential was 0.85 V, which is as high as observed in our previous study, due to the trifluoromethyl groups [50]. From this voltammetry profile, the highest occupied molecular orbital (HOMO) and lowest unoccupied molecular orbitals (LUMO) of PTPA were determined to examine the electrochemical oxidation stability [57]. The HOMO energy level was calculated to be −5.65 eV basis on the ferrocene standard level of 4.8 eV and the half-wave potential (E_1/2_) of ferrocene/ferrocenium ions. Such a low HOMO energy level of PTPA has the advantage of providing electrochemical oxidation stability against ambient oxygen and humidity [50]. Considering the HOMO level and the optical energy gap, derived from the UV/Vis results above, we also deduced a LUMO energy level of −2.92 eV.

### 3.2. Electrochemical Investigation

To evaluate the electrochemical performance of the PTPA, as a cathode in lithium ion batteries, we fabricated a CR 2032 coin-type cell with a lithium metal counter electrode. The coating on Al current collector of an organic cathode consisting of PTPA, PVDF, and Super-P (weight ratio was 5:1:4) was prepared. As shown in Appendix A, we observed the sphere-like structure of the PTPA, with an average size of 20 nm. Moreover, elemental mapping revealed the uniform distribution of C, O, N, and F elements over the entire electrode. The redox behavior of the cells was initially investigated by CV analysis, within the voltage range of 2.5–4.2 V, at a scan rate of 0.1 mV s^−1^. As displayed in Figure 6a, the cells exhibited two pairs of cathodic peaks (3.8–4.0 V and 4.0–4.2 V) and anodic peaks (3.8–3.9 V and 4.0–4.1 V). We considered that nitrogen atoms from the triphenylamine could act as the redox sites for lithium storage. As displayed in Figure 6a, the cells exhibited two pairs of cathodic peaks (3.8–4.0 V and 4.0–4.2 V) and anodic peaks (3.8–3.9 V and 4.0–4.1 V). The well-defined redox peaks between 4.0–4.2 V were attributed to the electrochemical reactions of the lithium insertion/extraction process of the triphenylamine moieties during the charge/discharge process, while the relatively weak redox peaks, occurring at 3.8–4.0 V, correspond to the organic ligands from the triphenylamine [36,58]. After the first activation cycle, the following CV curves were observed to be highly overlaid, suggesting that the cells were able to maintain electrochemical reversibility with low polarization.

Subsequently, the rate capability of the cells was measured at different current densities, of 20, 50, 100, 300, 500, and a recovery step at a current density of 20 mA g^−1^. As presented in Figure 6b, the specific capacity of the cells progressively decreased as the current density increased. The specific capacities were calculated to be 99.6, 87.8, 81.4, 73.2, and 58.3 mA h g^−1^ at current densities of 20, 50, 100, 300, and 500 mA g^−1^, respectively. After returning to the mild condition of 20 mA g^−1^, the specific capacities were recovered, very close to the initial specific capacity values, again confirming high reversibility and stability during charging and discharging processes.

The long-term cycling performances of the cell were also examined by evaluating 100 cycles at a current density of 50 mA g^−1^, as shown in Figure 6c. As expected, the cells retained stable cycling behavior without a significant decay of specific capacity. The specific capacity achieved after 100 cycles was 69.7 mA h g^−1^, which was on a par with the initial value, except for the activation cycles at the beginning of measurement.

A further manifestation of the stable cycling performances was obtained from the galvanostatic charge-discharge (GCD) profiles at the 5th, 10th, 20th, and 30th cycle. As shown in Figure 6d, no obvious change was observed in the GCD profiles during cycling; only small electrode polarization was detected with increasing cycle number, demonstrating the high electrochemical stability and reversibility of the cells. Based on these results, we believe that the new design of hyperbranched polymers could widen the selection of cathode materials for LIBs, which may hold the key for further improvements in the electrochemical performance of the energy storage technology.

## 4. Conclusions

A new arylene-ether type hyperbranched poly(triphenylamine) was prepared by S_N_Ar reaction. The success of the polymerization process depended on the reactivity of the monomers, and monomer 5, which had one hydroxyl group, was successfully polymerized to PTPA. The properties of the PTPA were characterized by NMR and IR spectra, SEC, TGA and DSC thermograms, UV spectra, and cyclovoltammetry curves. Lithium batteries that were fabricated with the PTPA as a cathode material demonstrated excellent electrochemical reversibility and stability at various current densities. Moreover, the cells continued to maintain stable cycling behavior over repeated cycling, suggesting opportunities of the PTPA, for applications as cathode materials, in lithium batteries. However, the molecular weight of the active material increased due to the CF_3_ groups introduced for polymerization, but it showed a lower specific capacity compared to previous studies. To improve the battery performance, we are conducting research on end-modification of the hyperbranched polymer with electroactive moieties and the design of new monomers that can reduce the molecular weight of repeating units.

## Data Availability

The data used in this study are available from the corresponding authors upon request.

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
