# Peer review of "Synthesis of Arylene Ether-Type Hyperbranched Poly(triphenylamine) for Lithium Battery Cathodes"

_materials, 2021, doi:10.3390/ma14247885_

Round 1
Reviewer 1 Report
In this manuscript, the authors synthesized a new poly(triphenylamine) having a hyperbranched structure and investigated the electrochemical performance as cathode materials for LIBs. This polymer exhibits comparative lithium storage capacity and cycling stability. However, some concerns need to be addressed before its acceptance for publication in Materials:
- In introduction section, the author claims that hyperbranched polymer features a layer surface area. Please provide the surface area of as-obtained samples.
- The detailed surface chemistry of the hyperbranched polymers should be analyzed using X-ray photoelectron spectroscopy.
- The morphology of hyperbranched polymers should be implemented in this work.
- What is the mass of active materials?
- Comparison between the hyperbranched polymers and other reported polymers cathodes should be provided.
- More recent works about metal ion batteries are suggested to be included: Angew. Chem. Int. Ed. 2020, 59, 17504-17510; J. Mater. Chem. A 2021, 9, 11879-11907; Adv. Energy Mater. 2017, 7, 1602880; J. Mater. Chem. A 2021, 9, 6402-6412.
Author Response
Please, see the response letter attached.

Reviewer 2 Report
- 62:“Here we report a new type of hyperbranched poly(triphenylamine) for a LIB cathode. The poly(triphenylamine) synthesized inthis study consists of triphenyamine and”, “Space” should be “inthis”.
- 107: Pt wire was used as the counter electrode, silver/silver nitrate (Ag/AgNO3) as the reference electrode The counter electrode should be Pt disk to avoid side reaction. If Pt wire was used as the counter electrode, excessive electric current density may result in electrolyte decomposition. So the counter electrode I prefer Pt disk. If the electrolyte composition remains stable, Pt wire is also OK.
Author Response

(The authors gave the same response as above.)

Reviewer 3 Report
Based on thermal, optical and electro-19 chemical analyses, a composite poly(triphenylamine) electrode was made in this paper. Overall, the research content of the manuscript is relatively new, and the process from theoretical analysis to experimental verification is reasonable. However, the following points need to be improved:
(1) The analysis of the current research situation is not thorough. It is necessary to pay attention to recent research results and to show the differences and innovations of the research in this article through citation and comparative analysis.
(2) The illustration of the article is not very clear, it is recommended to use a vector diagram.
(3) The conclusion needs to describe the deficiencies of the research in this article, and look forward to what needs to be paid attention to if the research is continued in the future.
Author Response

(The authors gave the same response as above.)

Reviewer 4 Report
The authors have synthesized a new poly(triphenylamine) having a hyperbranched structure and employed it in lithium ion batteries as an organic cathode material. It could be accepted for publication with a major revision. Specific comments are listed as follows.
-
- Please state the novelty of this material clearly.
- It is suggested to include SEM images to show its morphology.
- Please explain the reasons choosing the voltage range of 2.5 - 4.2 V in Fig. 6.
- Please include the efficiencies during the charging/discharging cycles.
- It is suggested to explain the mechanism of Li conduction or storage.
Author Response

(The authors gave the same response as above.)

Round 2
Reviewer 1 Report
The revised paper is ready for acceptance.
Reviewer 4 Report
The comments are well addressed. The current version is suitable for publication.